

# Murine models of colorectal cancer: the azoxymethane (AOM)/dextran sulfate sodium (DSS) model of colitis-associated cancer

Dzhuliia Dzhalilova[1], Natalia Zolotova[1], Nikolai Fokichev[2] and Olga Makarova[1]

[1] Avtsyn Research Institute of Human Morphology, Petrovsky National Research Centre of Surgery, Moscow, Russia

[2] Biological Department, Lomonosov Moscow State University, Moscow, Russian Federation

## ABSTRACT

**Background**. Colorectal cancer (CRC) is the third most common cancer. It is a heterogeneous disease, including both hereditary and sporadic types of tumors. CRC results from complex interactions between various genetic and environmental factors. Inflammatory bowel disease is an important risk factor for developing CRC. Despite growing understanding of the CRC biology, preclinical models are still needed to investigate the etiology and pathogenesis of the disease, as well as to find new methods of treatment and prevention.

**Objectives**. The purpose of this review is to describe existing murine models of CRC with a focus on the models of colitis-associated CRC. This manuscript could be relevant for experimental biologists and oncologists.

**Methodology**. We checked PubMed and Google from 01/2018 to 05/2023 for reviews of CRC models. In addition, we searched PubMed from 01/2022 to 01/2023 for articles using the azoxymethane (AOM)/dextran sulfate sodium (DSS) CRC model.

**Results**. Existing murine models of CRC include spontaneous, genetically engineered, transplantation, and chemically induced models. For the study of colitis-associated cancer (CAC), the AOM/DSS model is predominantly used. This model is very similar in histological and molecular characteristics to the human CAC, and is highly reproducible, inexpensive, and easy to use. Despite its popularity, the AOM/DSS model is not standardized, which makes it difficult to analyze and compare data from different studies.

**Conclusions**. Each model demonstrates particular advantages and disadvantages, and allows to reproduce different subtypes or aspects of the pathogenesis of CRC.

## INTRODUCTION

Among malignant neoplasms, according to GLOBOCAN worldwide 2020 data, colorectal cancer (CRC) is the third most common (10% of new cases of cancer) and second in the structure of mortality (9.4% of deaths from cancer). CRC is more common among men than among women. Age-standardized incidence rate of CRC (ASR—age standardized

Corresponding author
Dzhuliia Dzhalilova,
juliajal93@mail.ru

incidence rates) in men is 23.4 per 100,000 population, and in women 16.2, and the mortality rate (age-standardized mortality rates) is 11.0 and 7.2 per 100,000 population, respectively (*Hossain et al., 2022*).

About 3% of CRC cases are due to one of two autosomal dominant hereditary diseases: hereditary non-polyposis colorectal cancer (Lynch syndrome) and familial adenomatous polyposis. Approximately 90% of CRC cases are sporadic and occur in patients without genetic predisposition or family history of CRC (*Bogaert & Prenen, 2014*; *Lynch, Snyder & Shaw, 2015*; *Nascimento-Gonçalves et al., 2021*; *Li et al., 2022a*).

The colon is considered to be the organ with an inherently high rate of stem cell division, which leads to high risk of developing cancer throughout the life under the environmental and hereditary factors influence. According to existing concepts, CRC occurs as a result of a progressive accumulation of genetic and epigenetic changes that lead to the normal colon mucosa transformation into adenocarcinoma (*Barker, Ridgway & Van Es, 2009*; *Hossain et al., 2022*).

Comprehensive genomic analysis demonstrated that CRC is characterized by molecular heterogeneity, its individual cases are unique and contain, on average, 76 non-silent mutations each (*Wood et al., 2007*). Now there are four consensus molecular subtypes (CMS—consensus molecular subtypes) of CRC (*Guinney, Dienstmann & Wang, 2015*). CMS1 (MSI-immune type) accounts for 14% of CRC cases, is characterized by defects in the mismatch repair (MMR) system, which leads to microsatellite instability (MSI) and a high mutation rate. Often there are mutations in the genes *MSH6*, *RNF43*, *ATM*, *TGFBR2*, *BRAF*, *PTEN*, a high level of DNA methylation. In addition, this subtype is characterized by a pronounced infiltration of the tumor microenvironment by immune cells (mainly cytotoxic and activated Th1 lymphocytes), due to neoantigens resulting from hypermutability. CMS2, 3 and 4 subtypes demonstrate high degree of chromosomal instability (CIN) with the loss and/or acquisition of large chromosome regions, loss of heterozygosity and aneuploidy. They are characterized by an initial loss of the tumor suppressor gene *APC* followed by an activating mutation in *KRAS* and loss of *TP53*. Mutation frequency is significantly lower than in CMS1. 37% of CRC cases are CMS2 (canonical type). It is characterized by activation WNT and MYC signaling pathways, high expression of EGFR and HFN4A, and poor immunogenicity. CMS3 (metabolic subtype, 13% CRC) is characterized by dysregulation of a number of metabolic pathways, including glutamine, fatty acid, and lysophospholipid metabolism, and frequent *KRAS* mutations. In comparison to CMS2 and CMS4, CMS3 demonstrates lower level of somatic copy number alterations (SCNA) and higher microsatellite instability. CMS4 (mesenchymal subtype, 23% CRC) is characterized by the increased expression of genes regulating inflammation, matrix remodeling, stromal invasion, and angiogenesis. CMS4 tumors demonstrate extremely low hypermutation rates, microsatellite stable status, and very high SCNA levels. CMS4 tumors are located in the distal colon, infiltrated by Treg, Th17 and innate immune cells, and demonstrate high levels of cytokines IL-23 and IL-17 (*Guinney, Dienstmann & Wang, 2015*; *Thanki et al., 2017*; *Shah & Itzkowitz, 2022*).

The risk factors for CRC are genetic predisposition, high body mass index, lack of physical activity, tobacco and alcohol consumption, red meat reach diet, fried foods, low

dietary fiber, vitamin D deficiency, use of poor-quality drinking water, *etc.* (*Johnson et al., 2013*; *Chen et al., 2021*; *Farvid et al., 2021*).

## Colitis-associated CRC (CAC)

At the same time, one of the main risk factors for the development of CRC is inflammatory bowel disease (IBD) (*Shah & Itzkowitz, 2022*). IBD, including Crohn's disease (CD) and ulcerative colitis (UC), are chronic relapsing bowel diseases of the unknown etiology. IBD is characterized by indigestion and inflammation in the gastrointestinal tract. The main symptoms of IBD include diarrhea, abdominal pain, intestinal bleeding, and weight loss. IBD is often diagnosed in patients between 15 and 35 years of age. The incidence of IBD is increasing worldwide (*Seyedian, Nokhostin & Malamir, 2019*). Age-standardized prevalence increased from 79.5 (75.9–83.5) per 100,000 population in 1990 to 84.3 (79.2–89.9) per 100,000 population in 2017 (*GBD 2017 Inflammatory Bowel Disease Collaborators, 2020*). Due to unclear etiology and insufficient understanding of IBD pathogenesis, methods for prevention, diagnosis and treatment are not effective enough. In fact, modern methods of treatment can only relieve the exacerbation of inflammation, but not cure the disease (*Seyedian, Nokhostin & Malamir, 2019*). Chronic inflammation causes oxidative stress and DNA damage, which leads to the activation of tumor promoting genes and the inactivation of tumor suppressor genes. The result is a sequence of events that cause genetic (*e.g.*, mutations) and epigenetic (*e.g.*, methylation) changes, followed by clonal expansion of somatic epithelial cells. In contrast to the sporadic form of CRC characterized by dysplastic changes (adenomas, serrated polyps) developing as discrete small lesions, IBD is characterized by extensive areas of chronically inflamed mucosa are prone to neoplastic transformation. This phenomenon was called "field cancerization" (*Shah & Itzkowitz, 2022*). According to *Eaden, Abrams & Mayberry (2001)* the prevalence of CRC in UC patients is 3.2%, with a cumulative rate of 2%, 8%, and 18% CRC risk at 10, 20, and 30 years of illness, respectively. According to *Lutgens et al. (2013)* the risk of developing CRC on the background of IBD is 1%, 2% and 5% after 10, 20 and more than 20 years of illness, respectively. *Selinger et al. (2014)* estimated these rates at 1%, 3% and 7% after 10, 20 and 30 years of IBD. Meta-analysis of 31,287 UC patients from 44 studies by *Bopanna et al. (2017)*, demonstrated the overall prevalence of CRC in UC of 0.85% with a cumulative risk of 0.02%, 4.8% and 13.9% after 10, 20 and 30 years of the course of the disease.

Chronic inflammation leads to the development of a microenvironment enriched with immune cells that produce pro-inflammatory cytokines and growth factors and simultaneously increase local levels of reactive oxygen species. Consequently, cell proliferation and the risk of DNA damage increase. In case of a prolonged inflammatory response, tumor cell transformation occurs at a high frequency (*Stastna et al., 2019*). Colitis-associated colorectal cancer (CAC) is generally characterized by the same pathways that cause sporadic CRC (sCRC): chromosomal instability (CIN), microsatellite instability (MSI), and CpG island methylator phenotype (CIMP). Accordingly, mutations of the same driver genes are observed in CAC as in sCRC: *APC*, *KRAS*, *TP53*, *PIK3CA*, *SMAD4*, *ARID1A*, *MYC*, *etc.*, but the time and frequency of their occurrence can differ. CAC reflects the "Big Bang" model of evolution, characterized by chronic inflammation causing rapidly,

occurring molecular changes leading to neoplasia, rather than the gradual accumulation of lesions as in sporadic forms of CRC. There are no exact data on the position of CAC in the concept of the CMS classification of CRC. It is demonstrated that CAC is not characterized by CMS2 type. In CAC, there is an increase in the number of CD4+ T cells and monocytes, decrease in HNF4 expression and overexpression of OSMR, a tendency to epithelial-mesenchymal transition, which indicates a shift towards CMS4 (mesenchymal subtype) (*Shah & Itzkowitz, 2022*).

However, specific mechanisms for the development of CRC are still unknown, which makes difficult the early diagnostics and treatment of disease. The etiology and pathogenesis study of CRC on clinical material is complicated, since it is often almost impossible to establish the initial stages of the disease, the exact course and the presence of concomitant pathology and the use of drugs change the clinical manifestations of the disease. Studies of biopsy material provide information only about colon mucosa changes from several local colon areas. In its turn, the surgical material is represented only by the most severe cases of the disease and its late stages. The development of new drugs and treatments for CRC requires preclinical trials in experimental models. The current experimental models of CRC in laboratory animals include spontaneous, genetically engineered, transplant and chemically-induced models (*Nascimento-Gonçalves et al., 2021*). Each model has its own advantages and disadvantages and allows modeling different types of CRC.

The purpose of this review is to describe existing murine models of CRC with a focus on the models of colitis-associated CRC. This review can help experimental biologists to choose the most optimal colorectal cancer model for their purposes, and it will help oncologists to extrapolate experimental data more correctly.

## Survey methodology

We first searched through PubMed and Google for reviews of CRC models. The query "colorectal cancer model" was used, selecting only reviews from the last 5 years that discussed all types of mouse and rat models. Seven reviews were analyzed (*De-Souza & Costa-Casagrande, 2018*; *Stastna et al., 2019*; *Oliveira et al., 2020*; *Bürtin, Mullins & Linnebacher, 2020*; *Nascimento-Gonçalves et al., 2021*; *Li et al., 2022a*; *Neto et al., 2023*). Based on these data, the most widely used and adequate murine model of CAC was chosen—azoxymethane (AOM)/dextran sulfate sodium (DSS).

Secondly, we searched PubMed for articles using the AOM/DSS model. The "AOM DSS" query was used.

*Inclusion criteria*. Publication date between 01/2022 and 01/2023, free full text, original studies, AOM/DSS CAC model reproduced.

*Exclusion criteria*. Publications with priority date to January 1st, 2022; *ex vivo/in vitro* studies; commentaries, summaries, reviews, editorials, and duplicate studies.

Overall 91 articles were analyzed. The methodology of reproducing the AOM/DSS CAC model and the histological characteristics of the tumors were evaluated.

# RESULTS

## Murine CRC models
### Spontaneous models

Spontaneous tumors in the intestine of laboratory mice and rats develop rarely. For example, C57BL/6J mice developed colon cancer at 1% of cases. In colon cancer-prone rats substrain WF-Osaka the incidence of colon carcinoma varied from 30 to 40%. Spontaneous models are rarely used due to unpredictability and low reproducibility (*Nascimento-Gonçalves et al., 2021*).

### Genetically engineered murine models (GEMMs)

Genetically engineered models of colorectal cancer make it possible to study the genetic predisposition to the development of colorectal cancer, interaction with environmental and modifying factors, tumor microenvironment, as well as systemic immune responses (*Bürtin, Mullins & Linnebacher, 2020*; *Nascimento-Gonçalves et al., 2021*).

According to *Ye et al. (2020) KRAS, KIT, PIK3CA, MET* and *EGFR* are among the most frequently mutated oncogenes, and *TP53, APC, CDKN2A, STK11* and *FBXW7* are the most frequently mutated tumor suppressor genes in patients with CRC. In GEMMs mutations in *Apc*, *p53*, *K-ras* genes and DNA mismatch repair genes (MMR) are the most oftenly used, but other genetic mutations are also taken (*De-Souza & Costa-Casagrande, 2018*; *Stastna et al., 2019*; *Oliveira et al., 2020*; *Bürtin, Mullins & Linnebacher, 2020*; *Nascimento-Gonçalves et al., 2021*; *Li et al., 2022a*; *Neto et al., 2023*).

**Apc** The first genetic model for CRC was described in 1990. Min (Multiple intestinal neoplasia) is an ethylnitrosourea (Enu)-induced mutation in the murine *Apc* (adenomatous polyposis coli) gene (*Moser, Pitot & Dove, 1990*). The line of mice was named *Apc*[Min]. The acronym "min" means multiple intestinal neoplasms, and this is an autosomal dominant mutation, which in homozygous conditions is lethal to animals. Animals that are heterozygous for the mutation develop important anemic conditions at 60 days of life and develop tumors in the large and small intestine. As in familial adenomatous polyposis cases, *Apc*[Min] animals also develop colorectal adenomas, but they die at 120 days of life (*De-Souza & Costa-Casagrande, 2018*). *APC* encodes a key negative regulator of the Wnt-signaling pathway and it represents the most frequently mutated gene in CRC. Approximately 90% of sporadic colorectal tumors carry a mutation in *APC*. The *APC* locus was discovered through studying a rare hereditary syndrome, familial adenomatous polyposis (FAP). As the sequence identity of the human and mouse APC proteins exceed 89%, the mouse represents a suitable model to study the involvement of APC truncations in intestinal cancer (*Stastna et al., 2019*). The *Apc*[Min] mouse model is the only animal model of cancer that contains a single genetic alteration capable of producing a fully penetrating, consistent, and organ-specific tumor phenotype (*Nascimento-Gonçalves et al., 2021*). Throughout aging, there was a high mortality of *Apc*[Min] animals because of intestinal obstruction and anemia. A significant part of animals died before the tumor progresses to invasive carcinoma (*Bürtin, Mullins & Linnebacher, 2020*). Moreover, the polyps predominantly develop in the small intestine, and to a way lesser extent in the

colon, at the same time it is very rare for people to develop tumors of the small intestine (*Stastna et al., 2019*).

Mouse strains with other targeted genetic modifications at different locations of the *Apc* gene, such as $Apc^{\text{Min/850}}$, $Apc^{\Delta 716}$, $Apc^{1638N}$, $Apc1^{638T}$, $Apc^{\Delta 468}$ and $Apc^{\Delta 474}$, were developed to allow reproduction of disease models closer to CRC in humans and study the role of certain regions of the *APC* gene in the development of cancer. Transgenic mice with different mutations of the *APC* gene differ in the number of tumors and their morphology (*Stastna et al., 2019*; *Bürtin, Mullins & Linnebacher, 2020*; *Nascimento-Gonçalves et al., 2021*). Also rat model with *Apc* mutation created: Pirc rats. They developed adenomas similar to those found in humans, demonstrated the same progression to invasive carcinomas, and dependence on gender was observed, with males more prone to develop tumors in the intestinal tract than female rats (*Nascimento-Gonçalves et al., 2021*).

**DNA MMR** The most common form of hereditary CRC in humans is hereditary non-polyposis colon cancer (HNPCC) or Lynch syndrome. It is caused by autosomal dominant mutations in the DNA mismatch repair genes such as *MLH1*, *MSH2*, *MSH6*, and *PMS2*, that leads to the development of various cancers, including colorectal cancer. Developed tumors are not specific to the colon-rectum; they occur in other organs such as skin, lungs, lymphatic system, stomach, and small intestine (*Nascimento-Gonçalves et al., 2021*). Inactivation of DNA MMR genes leads to the development of microsatellite instability and hypermutative tumor phenotype. Microsatellite instability is also detected in some cases of spontaneous CRC (*Bürtin, Mullins & Linnebacher, 2020*). Therefore, a number of mouse models were developed with knockouts MMR genes such as *Mlh1*, *Mlh3*, *Msh2*, *Msh6*, and *Pms2*, which is in details described in the *Lee, Tosti & Edelmann (2016)*.

**p53** Mutations in the p53 gene are found in 60% of human CRC (*Neto et al., 2023*). Animals with *p53* knockout rarely develop or don't develop colorectal tumors (*De-Souza & Costa-Casagrande, 2018*). Surprisingly, given the proposed role for loss of function mutations of the *P53* gene in the development of human colorectal cancer, *Clarke, Cummings & Harrison (1995)* found no evidence for either an increase in the rate of adenoma formation in $Apc^{+/-}\ p53^{-/-}$ animals, or an increased rate of progression to malignancy compared with $Apc^{+/-}\ p53^{+/+}$ mice. The role of mutations in *p53* in the development of colon cancer is being actively studied, and the suppression of the mutant *p53* function *via* the inhibition of nuclear accumulation is expected to be an effective strategy against malignant progression of colorectal cancer (*Nakayama & Oshima, 2019*).

**KRAS** *KRAS* belonging to the RAS family, is one of the most prominent proto-oncogenes, and associated with various oncogenic pathways including PI3K/AKT/mTOR signaling to promote proliferation and to suppress apoptosis of tumor cells. *KRAS* mutation found in more than 40% of patients with CRC (*Li et al., 2022a*). Mice with mutations in codon 12 of the *K-ras* gene ($K\text{-}ras^{\text{G12D}}$) have regions of hyperplasia in the colon as well as aberrant crypt and ring cells. Associating the $K\text{-}ras^{\text{G12D}}$ and $Apc^{\text{Min}}$ mutations lead to increase in the number of lesions in the colon, as well as the presence of completely undifferentiated cells. Another mutation, $K\text{-}ras^{\text{v12}}$, alone is not capable of inducing tumorigenesis, but once it is associated with mutations in *Msh2* gene, it promotes a greater number of tumors in

the small and in the large intestine, than $Msh2^{-/-}$ animals (*De-Souza & Costa-Casagrande, 2018*).

Thus, genetically engineered models made a huge contribution to understanding the molecular mechanisms of CRC initiation and progression. However, these models include a number of limitations. Firstly, cancer development is a step-by-step process with initial driver mutation and subsequent acquisition of further mutations, and therefore this process can only be partially captured in mouse tumor models by combination of mutation. Secondly, the number of combined mutations in mice is limited, as the resulting phenotype often demonstrates a dramatic lifespan reduction. In addition, creating genetically engineered models is time consuming and expensive, as breeding transgenic mice often takes several generations and requires careful breeding to produce the desired changes. With regard to animal welfare, it should be noted that during the breeding process, many "rejected" mice appear, which are not used either for further breeding or for the research purposes. Furthermore, the constructing technique for a transgene or viral vector requires special skills. Although genetically engineered models are a valuable tool for fundamental research, their use for preclinical studies is limited due to the lack of genetic heterogeneity, on the one hand, and inconsistencies with the mechanisms of human tumor development, on the other (*Bürtin, Mullins & Linnebacher, 2020*).

### Transplant models

Transplant models are performed by transplanting a tumor from one living organism to another. Transplant models are classified according to three main parameters:

(1) Type of transplant. Either tumor cell suspensions or tumor tissue fragments are used. It is easier to work with a cell suspension, cheaper, and the tumor engraftment rate is higher. Cell lines of tumors are phenotypically and genetically quite homogeneous and do not reproduce the heterogeneity of real tumors; in addition, tumor cells due to *in vitro* selection usually tend to be poorly differentiated and developing tumors are more aggressive in comparison to human CRC. Tumor tissue fragments make it possible to preserve the initial tumor microenvironment, genetic and molecular heterogeneity of tumor cells, and are more adequate model, but their engraftment rate is lower.

(2) Source of the graft. Syngeneic tumor transplantation is characterized by the engraftment of tumor tissue or a tumor cell line within the same animal line; xenogeneic transplants are obtained from other animal lines or human donors. Patient-derived xenografts retain the pathological and molecular characteristics of individual patient CRC and are therefore the most reliable for preclinical drug development. However, in xenograft models, tumor cells or tumor fragments are implanted in immunocompromised animals, which makes it impossible to study the role of immune responses in carcinogenesis.

(3) Site of the transplantation. The implantation of tumor could be various: directly into the colon or rectum (orthotopic models) or subcutaneously, into the renal capsule, spleen, bloodstream (heterotopic models). The most widely used method in case to initiate reproduction, provide access and fast tumor growth is subcutaneous inoculation, nonetheless, the tumor microenvironment is different from the colon and metastasis do not develop. In comparison to heterotopic models the orthotopic model, are technically more

difficult to create, requiring the use of imaging techniques (*e.g.*, ultrasound) to control cell implantation and animal anesthesia.

Transplant models make it possible to reproduce tumor invasion, cancer progression to advanced stages, and metastasis to other organs. But they are not suitable for studying the induction and early stages of carcinogenesis, as well as studying the role of the immune system and inflammation in the pathogenesis of CRC (*De-Souza & Costa-Casagrande, 2018*; *Stastna et al., 2019*; *Oliveira et al., 2020*; *Bürtin, Mullins & Linnebacher, 2020*; *Nascimento-Gonçalves et al., 2021*; *Neto et al., 2023*).

### Chemically-induced CRC models (CIMs)

Carcinogen-induced CRC in mice is a rapid, relatively inexpensive, highly reproducible model that mimics the human adenoma-to-adenocarcinoma progression sequence.

At the moment, a wide range of chemically induced CRC models are available. The most commonly used groups of carcinogens are:

(1) heterocyclic amines (HCAs) such as 2-amino-3-methylimidazo[4,5-f]quinoline (IQ) and 2-amino-1-methyl-6-phenylimidazo [4,5-b] pyridine (PhIP);

(2) aromatic amines such as 3,2-dimethyl-4-aminobiphenyl (DMAB);

(3) alkylnitrosamide compounds such as N-methyl-N-nitro-N-nitrosoguanidine (MNNG) and methylnitrosourea (MNU);

(4) 1,2-dimethylhydrazine (DMH), azoxymethane (AOM), methylazoxymethanol (MAM).

The introduction of chemical carcinogens to animals is possible with the free access to water and food or through a gastric tube, enema, or intraperitoneal or subcutaneous injection (*Stastna et al., 2019*; *Li et al., 2022a*).

**Heterocyclic amines**

Heterocyclic amines are among the most common genotoxic mutagens present in the environment, and the human body is exposed to them in daily life. Heterocyclic amines are formed in food when amino acids and proteins are heated. Throughout frying meat and fish, various imidazoquinoline, imidazoquinoxaline and imidazopyridine compounds are formed, which have a strong mutagenic effect.

Among heterocyclic amines, 2-amino-3-methylimidazo[4,5-f]quinolone (IQ) and 2-amino-1-methyl-6-phenylimidazo[4,5-b]pyridine (PhIP) are used to induce CRC in rodents. Rats given long-term IQ or PhIP develop tumors in the small and large intestine, mammary gland, and prostate. However, the incidence of colon tumors is low and ranges from 5% to 28% when these drugs are administered with food for up to 1 year. When adding 100 p.p.m. PhIP in rat diets takes approximately 2 years to reach a 50% incidence of colon cancer, and at 25 p.p.m. PhIP after 2 years no colon carcinomas were observed. It should be stated that the combination of PhIP with a high-fat diet leads to accelerated tumor formation.

Thus, the model of CRC induced by heterocyclic amines is applicable for the development of methods for preventing the development of tumors under the action of food carcinogens. The disadvantages of this model are the relatively low incidence of tumor development

and too long time of their development (*De-Souza & Costa-Casagrande, 2018*; *Stastna et al., 2019*; *Oliveira et al., 2020*; *Neto et al., 2023*).

**Aromatic amines**

*Lorenz & Stewart (1941)* initially observed the chemical induction of intestinal tumors in mice treated with polyaromatic hydrocarbons—dibenzanthracene or methylcholanthrene. DMAB shares structural similarities with mutagens found in well-done meat. The carcinogenic effect of DMAB (3, 2′-dimethyl-4-aminobiphenyl) was first described in 1950th by *Walpole, Williams & Roberts*. The researchers established the induction of colon tumors in rats by DMAB subcutaneous administration.

Subcutaneous weekly administration of DMAB at 50 mg/kg body weight in male F344 rats provided multiple colon tumors in 27% animals fed low and 75% animals fed high fat diet, respectively (*Doi et al., 2007*). DMAB contributes to both adenomas and adenocarcinomas epithelial neoplasms with a multiplicity of 1.2–2.7 tumor nodes per animal.

One of the key shortcommings of this model is that it suggests multiple injections of DMAB to induce colon tumors. On a molar basis, DMAB is considered to be less effective than AOM or DMH in rodent models. Another shortcoming of this model is the initiation of neoplasms in other organs: mammary adenocarcinomas in female rats, salivary gland sarcomas, squamous cell carcinomas of the ear canal and skin, squamous cell papillomas of the fundus of the stomach (forestomach), sarcomas and lymphomas, and bladder urothelial carcinomas (*Nascimento-Gonçalves et al., 2021*; *Neto et al., 2023*).

**Alkyl nitrosamide substances**

Methylnitrosourea (MNU) and N-methyl-N'-nitro-N-nitrosoguanidine (MNNG) are direct alkylating agents that do not require metabolic activation and thus are potent local carcinogens.

Intrarectal injection of MNU at 1–3 mg per rat weekly for 5 month provided colon tumors in 100% of male F344 rats, of which 43% of the tumors were adenocarcinomas and 57% were adenomas. All neoplasms were detected in rectum and distal colon, where MNU and MNNG were injected.

When using alkylnitrosamide substances in rats and mice, most of the induced colon tumors were sessile or polypoid lesions that were well differentiated and often invaded the submucosa. Metastases are usually not detected.

Since biochemical activation is not necessary, these carcinogens are perfect for initiation tumors of colon in animals and investigation the changing effects of xenobiotics without establishing the metabolism of the initiating carcinogen. MNU and MNNG administered intrarectally selectively initiate tumors in the rectum and distal colon, such models are widely spread.

The main disadvantage of this model is that intrarectal injection is a significant technical problem, the reproducibility of such experiments depends on the skills of the experimenter, and the quantification of carcinogens administered intrarectally is difficult (*Bürtin, Mullins & Linnebacher, 2020*; *Nascimento-Gonçalves et al., 2021*; *Neto et al., 2023*).

**DMH, AOM, MAM**

The most commonly used chemicals to induce CRC are 1,2-dimethylhydrazine (DMH) and its metabolite azoxymethane (AOM). They are potent carcinogens, causing a wide range of mutations in key genes that code for components of multiple intracellular signaling cascades (*De-Souza & Costa-Casagrande, 2018*; *Stastna et al., 2019*; *Bürtin, Mullins & Linnebacher, 2020*; *Nascimento-Gonçalves et al., 2021*; *Li et al., 2022a*; *Neto et al., 2023*).

DMH is oxidized in the liver to azomethane, which is then oxidized to AOM and then hydroxylated to methylazoxymethanol (MAM). MAM enters the intestine with bile or through the bloodstream in the form of glucuronides and glucosides.

MAM is cleaved by the effect of colon and liver enzymes to form formaldehyde and a strong alkylating agent,—the methyldiazonium ion. DNA alkylation results in base mismatch and mutagenesis. Thus, guanine is methylated in DNA at the N-7 position. Alkylated guanine combines with thymidine instead of cytosine. In the course of further replication, a nucleotide replacement occurs (*Venkatachalam et al., 2020*).

DMH is typically administered subcutaneously, but intraperitoneal and intrarectal administration were also stated in the literature. Dosing, number of injections and duration of the experiment vary significantly: 2–200 mg/kg body weight, 1–30 injections, 2–20 weeks (*Venkatachalam et al., 2020*). Protocol proposed by *Gurley, Moser & Kemp (2015)*, suggests administering 15 $\mu$g of DMH per gram of body weight to mice subcutaneously once a week for 12 weeks.

AOM is administered, as a rule, once a week intraperitoneally at a dose of 10-15 mg/kg of animal body weight for 2-10 weeks, the development of tumors is assessed after 16-36 weeks from the start of the experiment (*Waly et al., 2014*; *Whetstone et al., 2016*; *Velázquez et al., 2016*; *Uyar et al., 2022*). According to *Whetstone et al. (2016)* in male Balb/c mice, which were intraperitoneally injected with 10 mg/kg of AOM once a week for 6 weeks, 30 weeks after the first injection of AOM, adenomas were observed in 63% of the animals, and adenocarcinomas—in 21%.

The introduction of DMH or AOM leads to the development of epithelial neoplasia, which begins with the developing of abnormal crypts in the colon—the so-called aberrant crypt foci (ACF); ACF further progresses to adenoma and then to malignant adenocarcinoma (*Stastna et al., 2019*).

Different mice strains vary according to the AOM sensitivity. A/J and SWR/J mice are highly sensitive to AOM with high incidence of colon tumors. C57B/L6 and Balb/c mice are moderately sensitive with a relatively lower incidence of colon tumors in comparison to A/J and SWR/J mice, while administration of AOM to AKR/J and 129/SV mice does not induce colon tumors (*Rosenberg, Giardina & Tanaka, 2009*).

The disadvantages of this model are the long tumor induction period, not very high reproducibility potential, and the need for multiple injections of carcinogen.

## AOM/DSS model of colitis-associated colorectal cancer

As was mentioned above, in humans, the development of CRC is often associated with IBD. Therefore, *Tanaka et al. (2003)* proposed a new model of colorectal tumors associated with chronic inflammation. Male ICR mice were injected intraperitoneally with AOM (10

mg/kg body weight) once, and then, one week after the injection, the water in the drinkers was replaced with a 2% solution of dextran sulfate sodium (DSS) for 7 days. DSS is a non-genotoxic pro-inflammatory agent used to model acute and chronic colitis in rodents. By the 12th week of the experiment, all animals developed colon adenocarcinomas.

This model proved to be very convenient, relatively inexpensive, and highly reproducible, and is widely used in studies of colitis-associated carcinogenesis. Investigations made on this model reflected the significance of the inflammation process in CRC development and revealed some of the mechanisms of inflammation-related colon carcinogenesis in the gut, with an emphasis on the function of pro- and anti-inflammatory cytokines (*Tanaka et al., 2003*; *Greten et al., 2004*; *Tang et al., 2012*). This model is actively used in the search for therapeutic agents for anticancer therapy (*Cai et al., 2022*; *Chen et al., 2022c*; *Deng et al., 2022*; *Kim et al., 2022*; *Leung, Lo & El-Nezami, 2022*; *Lin et al., 2020*; *Luo et al., 2022a*; *Luo et al., 2022b*; *Pan et al., 2022*).

Histologically, dysplasia in human IBD often looks like dysplasia in sporadic adenomas, resembling tubular, tubulovillous or villous adenomas. Although a clear distinction cannot be relied upon, dysplastic cells in sporadic adenomas often occupy the upper portion of neoplastic crypts (so-called, top-down dysplasia) whereas with colitis, the dysplastic cells tend to occupy the entire height of the crypts (*Shah & Itzkowitz, 2022*). In general, tumors developed in AOM/DSS-treated mice are similar to human CAC. They are located in the same medial and distal colon, histologically characterized as tubular adenomas or moderately differentiated tubular adenocarcinomas. There is invasion of tumors into the submucosa, muscle and serous membranes. It is important to mention that, unlike tumors, caused only by AOM, which are mainly adenomas, exposure to AOM/DSS can induce the formation of a complete process of colon oncogenesis, progressing from the initial proliferation of crypts to the final development of colon carcinoma (*Li et al., 2022a*). However, due to different approaches to the induction of this model, it is difficult to compare the obtained results with each other.

As with AOM, different mice strains demonstrate different sensitivity to AOM/DSS. *Suzuki et al. (2006)* evaluated the development of colon tumors in four different strains of mice: Balb/c, C3H/HeN, C57BL/6N and DBA/2N. The animals were injected once intraperitoneally with 10 mg/kg of body weight AOM, then a week later the water in the drinkers was replaced with a 1% DSS solution for 4 days, the animals were withdrawn from the experiment 18 weeks after the injection of AOM. The incidence of colon adenocarcinoma in mice was 100% for Balb/c, 50% for C57BL/6N, and no tumors were found in the other two strains. At the same time, inflammatory changes in the colon were most pronounced in C3H/HeN mice, followed by Balb/c and DBA/2N, and inflammatory changes were the least pronounced in C57BL/6N mice.

Sometimes DMH is used instead of AOM, and instead of DSS, another pro-inflammatory agent is 2,4,6-trinitrobenzenesulfonic acid (TNBS). Sometimes DMN is used instead of AOM, and instead of DSS, another pro-inflammatory agent is TNBS (*Abad et al., 2003*; *Antoniou et al., 2016*). Thus, there are DMH/DSS, DMH/TNBS and AOM/TNBS modes of CAC (*Neto et al., 2023*).

Despite the widespread use of AOM/DSS-induced CAC, the model has not been standardized. We analyzed AOM/DSS CAC induction techniques in 91 free access original studies published in the public domain between 01/2022 and 01/2023 in the PubMed database (Table 1).

The predominant majority of studies (90 out of 91 articles) were performed on mice, mainly C57BL/6, C57BL/6J, C57BL/6N (70% of works) and Balb/c (19% of works). In 52% of cases, the study was carried out on male animals. CRC was induced by 1-5 injections of AOM at a total dose of 7.4-60 mg/kg of animal body weight and 1-4 cycles of consumption of 1–5% DSS solution for 2-8 days with breaks of 7-30 days. AOM was administered, as a rule, intraperitoneally (89 out of 91 articles). In 91% of cases there was only one AOM injection on day 1 of the experiment. The most commonly used AOM doses are 10 mg/kg (57% of articles) and 12.5 mg/kg (19% of articles). Mostly DSS is added to drinkers for three cycles (80% of articles), lasting 7 (70%) or 5 (21%) days at a concentration of 2% (42% of articles) or 2.5% (28% of articles). In 71% of articles, the breaks between DSS cycles were 14 days. The first cycle of DSS starts on days 1-8 of the experiment, in 51% of articles—on day 8. The duration of the experiment ranges from 6 to 25 weeks, most often 10 (21%), 11 (13%) or 12 (19%) weeks. Unfortunately, in some articles the molecular weight of the DSS used is not indicated, but most of the authors use DSS with a molecular weight of 36–50 kDa.

According to this data, the average scheme should look like stated below. Adult C57BL/6 mice on the 0th day of the experiment are injected intraperitoneally with AOM at a dose of 10 mg/kg of animal body weight, then 7 days leave them without exposure. Starting from the 8th, 29th and 50th days of the experiment, the water in the drinkers is replaced with 2% DSS for 1 week. Animals are withdrawn from the experiment at day 84 (Fig. 1).

Such experimental scheme is presented in four analyzed studies (*Hong et al., 2022*; *Pan et al., 2022*; *Chen et al., 2022c*; *Luo et al., 2022b*). In these studies, macroscopically visible tumor nodes of various sizes were found in the distal colon of all animals, with averagely about 6-10 tumor nodes per animal. Unfortunately, there is no qualitative histological description of tumors in the articles. Histological changes were characterized as deformed crypts, atypical epithelial hyperplasia, and polyps.

In general, in most of the works analyzed, there is no pathomorphological characterization of tumors induced by AOM/DSS. The observed changes are most often characterized as low-grade and high-grade dysplasia, adenomas or adenomatous polyposis. The term "adenocarcinoma" was used to describe AOM/DSS-induced tumors in only 20 out of 91 analyzed papers. The development of adenocarcinomas was noted at 8-25 weeks of the experiment (most often 12 weeks) with an injection of 10-30 mg/kg AOM (most often 12.5 mg/kg) and 1-4 cycles of consumption of 1.5–5% DSS during 2–7 days (most often 3 cycles of 2.5% DSS for 7 days). However, the incidence of adenocarcinomas was assessed only in two studies. According to *Bernardazzi et al. (2022)* in male C57BL/6 mice after a single intraperitoneal injection of AOM 12.5 mg/kg and 3 cycles of 2.5% DSS for 7 days to the 57th day of the experiment (8 weeks) 90% of the lesions were adenomas, and 10% were adenocarcinomas. According to *Zhang et al. (2022a)* study of high-grade dysplasia adenocarcinoma (including early carcinoma), 25% of female C57BL/6J mice

**Table 1  AOM/DSS CAC induction techniques.**

| Reference | Animals | AOM | DSS | Experiment duration |
|---|---|---|---|---|
| *Zhang et al. (2022d)* | Mice, sex?, Balb/C, 6–8 weeks | 1 injection i/p 10 mg/kg at day 0 | 3 7-days cycles 2.5% from days 8, 29, 50 | 57 days |
| ***Bernardazzi et al. (2022)*** | **Mice, male, C57BL/6, 7 weeks** | **1 injection i/p 12,5 mg/kg at day 0** | **3 7-days cycles 2.5% from days 8, 29, 50** | **57 days** |
| *Fragoso et al. (2022)* | Mice, female, Swiss Webster, 8 weeks | 1 injection i/p 7,4 mg/kg at day 0 | 3 7-days cycles 4% from days 8, 29, 50 | 70 days |
| *Moon et al. (2022)* | Mice, male, C57BL/6J, 8 weeks | 1 injection i/p 10 mg/kg at day 0 | 3 7-days cycles 1% from days 8, 29, 50 | 70 days |
| *Wu et al. (2022a)* | Mice, male, C57BL/6J, 6 weeks | 1 injection i/p 10 mg/kg at day 0 | 3 7-days cycles 1% from days 8, 29, 50 | 70 days |
| *Ghosh et al. (2022)* | Mice, male, C57BL/6, 7 weeks | 1 injection i/p 10 mg/kg at day 0 | 3 7-days cycles 1.5% from days 8, 29, 50 | 70 days |
| *Chen et al. (2022b)* | Mice, female, C57BL/6J, 7 weeks | 1 injection i/p 10 mg/kg at day 0 | 3 7-days cycles 2% from days 8, 29, 50 | 70 days |
| *Deng et al. (2022)* | Mice, male, C57BL/6, 7 weeks | 1 injection i/p 10 mg/kg at day 0 | 3 7-days cycles 2.5% from days 8, 29, 50 | 70 days |
| *Chen et al. (2022a)* | Mice, male, C57BL/6J, 8 weeks | 1 injection i/p 10 mg/kg at day 0 | 3 7-days cycles 2.5% from days 8, 29, 50 | 70 days |
| *Luo et al. (2022b)* | Mice, male, Balb/C, 5 weeks | 1 injection i/p 10 mg/kg at day 0 | 3 7-days cycles 2.5% from days 8, 29, 50 | 70 days |
| ***He et al. (2022)*** | **Mice, male, C57BL/6, 8–10 weeks** | **1 injection i/p 12 mg/kg at day 0** | **3 7-days cycles 1.5% from days 8, 29, 50** | **70 days** |
| *Martínez-Gregorio et al. (2022)* | Mice, male, Balb/C, 5–6 weeks | 1 injection i/p 12.5 mg/kg at day 0 | 3 7-days cycles 2% from days 8, 29, 50 | 70 days |
| *Liu et al. (2022e)* | Mice, male, C57BL/6J, 5 weeks | 1 injection i/p 10 mg/kg at day 0 | 3 7-days cycles 2.5% from days 8, 29, 50 | 77 days |
| *Yang et al. (2022b)* | Mice, male, Balb/C, 6 weeks | 1 injection i/p 10 mg/kg at day 0 | 3 7-days cycles 2.5% from days 8, 29, 50 | 77 days |
| *Hong et al. (2022)* | Mice, female, C57BL/6, 6 weeks | 1 injection i/p 10 mg/kg at day 0 | 3 7-days cycles 2 or 1% from days 8, 29, 50 | 85 days |
| *Chen et al. (2022c)* | Mice, female, C57BL/6, age? | 1 injection i/p 10 mg/kg at day 0 | 3 7-days cycles 2% from days 8, 29, 50 | 84 days |
| *Pan et al. (2022)* | Mice, male, C57BL/6J, age? | 1 injection i/p 10 mg/kg at day 0 | 3 7-days cycles 2.5% from days 8, 29, 50 | 85 days |
| *Luo et al. (2022a)* | Mice, sex?, C57BL/6, age? | 1 injection i/p 10 mg/kg at day 0 | 3 7-days cycles 2% from days 8, 29, 50 | 84 days |
| *Wang et al. (2022b)* | Mice, male, C57BL/6, 6 weeks | 1 injection i/p 10 mg/kg at day 0 | 3 7-days cycles 2.5% from days 8, 29, 50 | 84 days |
| *Xie et al. (2023)* | Mice, female, C57BL/6, 6–8 weeks | 1 injection i/p 12 mg/kg at day 0 | 3 7-days cycles 2% from days 8, 29, 50 | 84 days |
| ***Zhang et al. (2022a)*** | **Mice, female, C57BL/6J, 6 weeks** | **1 injection i/p 12 mg/kg at day 0** | **3 7-days cycles 2.5% from days 8, 29, 50** | **84 days** |
| *Zhou et al. (2022a)* | Mice, female, C57BL/6J, 7 weeks | 1 injection i/p 12 mg/kg at day 0 | 3 7-days cycles 2.5% from days 8, 29, 50 | 84 days |

**Table 1** (*continued*)

| Reference | Animals | AOM | DSS | Experiment duration |
|---|---|---|---|---|
| *Liu et al. (2022b)* | Mice, male, C57BL/6, 8–10 weeks | 1 injection i/p 10 mg/kg at day 0 | 3 7-days cycles 2% from days 8, 29, 50 | 87 days |
| *Wu et al. (2022b)* | Mice, male, C57BL/6N, 6–8 weeks | 1 injection i/p 10 mg/kg at day 0 | 3 7-days cycles 2% from days 8, 29, 50 | 91 days |
| *Liu et al. (2022g)* | Mice, male, C57BL/6J, 6–8 weeks | 1 injection i/p 12.5 mg/kg at day 0 | 3 7-days cycles 2.5% from days 8, 29, 50 | 91 days |
| *Jing et al. (2022)* | Mice, sex?, C57BL/6N, 8–10 weeks | 1 injection i/p 10 mg/kg at day 0 | 3 7-days cycles 2% from days 8, 29, 50 | 97 days |
| *Su et al. (2022)* | Mice, female, C57BL/6J, 4 weeks | 1 injection i/p 10 mg/kg at day 0 | 3 7-days cycles 2% from days 8, 29, 50 | 98 days |
| *Lin et al. (2022)* | Mice, male, C57BL/6, 6–8 weeks | 1 injection i/p 10 mg/kg at day 0 | 3 7-days cycles 3% from days 8, 29, 50 | 98 days |
| ***Tao et al. (2022)*** | **Mice, male, C57BL/6N, 6–8 weeks** | **1 injection i/p 12.5 mg/kg at day 0** | **3 7-days cycles 2.5% from days 8, 29, 50** | **98 days** |
| *Ren et al. (2022)* | Mice, male, C57BL/6, 6–8 weeks | 1 injection i/p 10 mg/kg at day 0 | 3 7-days cycles 2% from days 8, 29, 50 | 105 days |
| *Hases et al. (2022)* | Mice, male and female, C57BL/6J, 10 weeks | 1 injection i/p 10 mg/kg at day 0 | 3 7-days cycles 2.5% from days 8, 29, 50 | 63 or 105 days |
| *Liu et al. (2022c)* | Mice, female, C57BL/6, age? | 1 injection i/p 10 mg/kg at day 0 | 3 7-days cycles 2% from days 8, 29, 50 | 112 days |
| *Chao et al. (2022)* | Mice, male, C57BL/6, 8 weeks | 1 injection i/p 10 mg/kg at day 0 | 3 7-days cycles 2% from days 8, 29, 50 | 112 days |
| *Qin et al. (2022)* | Mice, female, Balb/C, 4–5 weeks | 1 injection i/p 10 mg/kg at day 0 | 3 7-days cycles 2% from days 8, 29, 50 | 91 or 147 days |
| *Zhu et al. (2022)* | Mice, female, C57BL/6, 8 weeks | 1 injection i/p 10 mg/kg at day 0 | 3 7-days cycles 2.5% from days 8, 29, 50 | 120 days |
| *Kennel et al. (2022)* | Mice, male and female, 129/Sv/Swiss, 8–12 weeks | 1 injection i/p 10 mg/kg at day 0 | 3 cycles 7, 7 and 5 days 2.5% from days 8, 29, 50 | 84 days |
| *Hiraishi et al. (2022)* | Mice, female, C57BL/6CrSlc, 6 weeks | 1 injection i/p 12 mg/kg at day 0 | 3 7-days cycles 2% from days 8, 22, 36 | 147 days |
| *Shao et al. (2022)* | Mice, male, Balb/C, 7 weeks | 1 injection i/p 10 mg/kg at day 0 | 3 7-days cycles 2.5% from days 8, 22, 36 | 91 days |
| *Leung, Lo & El-Nezami (2022)* | Mice, male, Balb/C, 7 weeks | 1 injection i/p 10 mg/kg at day 0 | 3 7-days cycles 2.5% from days 8, 22, 36 | 84 days |
| *Collard et al. (2023)* | Mice, male, Balb/C, 5 weeks | 1 injection i/p 10 mg/kg at day 0 | 3 cycles 7, 7 and 4 days 1.5% from days 8, 22, 43 | 84 days |
| *Yu et al. (2022)* | Mice, male, C57BL/6N, 6 weeks | 1 injection i/p 10 mg/kg at day 0 | 3 7-days cycles 2% from days 8, 43, 64 | 133 days |
| ***Schepelmann et al. (2022)*** | **Mice, female, Balb/C и C57Bl/6, 14 month (old)** | **1 injection i/p 12.5 mg/kg at day 0** | **3 5-days cycles (3% for C57Bl/6 and 2.5% for Balb/C) from days 8, 29, 50** | **84 days** |
| *Gong et al. (2022)* | Mice, male, C57BL/6J, 6–8 weeks | 1 injection i/p 12.5 mg/kg at day 0 | 3 5-days cycles 2.5% from days 8, 27, and 2% from 46 day | 65 days |

**Table 1** (*continued*)

| Reference | Animals | AOM | DSS | Experiment duration |
|---|---|---|---|---|
| *Ma et al. (2022)* | Mice, male, C57BL/6, 6 weeks | 1 injection i/p 12.5 mg/kg at day 0 | 3 5-days cycles 2.5% from days 8, 38, 58 | 85 days |
| **Chou et al. (2022)** | **Mice, male, C57BL/6, 7 weeks** | **1 injection i/p 12.5 mg/kg at day 0** | **3 5-days cycles 3.5% from days 8, 27, 46** | **70 days** |
| *Martínez-Gutierrez et al. (2022)* | Mice, male and female, C57BL/6J, 8–10 weeks | 1 injection i/p 12.5 mg/kg at day 0 | 3 5-days cycles 3% from days 8, 29, 50 | 70 days |
| *Zhou et al. (2022b)* | Mice, male, C57BL/6J, 7 weeks | 1 injection i/p 12.5 mg/kg at day 0 | 3 5-days cycles 3, 2.5 and 2,5% from days 8, 29, 50, respectively | 80 days |
| **Liu et al. (2022d)** | **Mice, male, C57BL/6, 5 weeks** | **1 injection i/p 15 mg/kg at day 0** | **3 2-days cycles 5% from days 8, 29, 50** | **112 days** |
| *Wang et al. (2022a)* | Mice, sex?, C57BL/6J, age? | 1 injection i/p 10 mg/kg at day 0 | 3 5-days cycles 3% from days 6, 25, 44 | 80 days |
| *Lee et al. (2022)* | Mice, male, line?, 6 weeks | 1 injection i/p 12.5 mg/kg at day 0 | 3 5-days cycles 2.5% from days 6, 25, 44 | 63 days |
| *Yang et al. (2022a)* | Mice, sex?, line?, 8–9 weeks | 1 injection i/p 10 mg/kg at day 0 | 3 5-days cycles 2% from days 6, 25, 44 | 63 days |
| *Mao et al. (2022)* | Mice, sex?, line?, 8–10 weeks | 1 injection i/p 10 mg/kg at day 0 | 3 5-days cycles 2.5% from days 6, 25, 44 | 80 days |
| **Moritsch et al. (2022)** | **Mice, sex?, C57BL/6, age?** | **1 injection i/p 12.5 mg/kg at day 0** | **3 5-days cycles 2.5%, 2.5% and 2% from days 6, 25, 44, respectively** | **79 days** |
| *Klingler et al. (2022)* | Mice, male, Rosa26-lacZ и Rosa26-mTmG, 6–8 weeks | 1 injection i/p 12.5 mg/kg at day 0 | 3 5-days cycles 2%, 1.5% and 1% from days 6, 25, 44, respectively | 90 days |
| *Sun et al. (2022)* | Mice, sex?, C57BL/6, age? | 1 injection i/p 10 mg/kg at day 0 | 3 5-days cycles 1.8% from days 6, 27, 48 | 70 days |
| *Sharma et al. (2022)* | Mice, male and female, C57BL/6J, age? | 1 injection i/p 10 mg/kg at day 0 | 3 6-days cycles 2.5%, 2% and 2% from days 6, 26, 46, respectively | 80 days |
| *Zhang et al. (2022e)* | Mice, female, C57BL/6J, 8–10 weeks | 1 injection i/p 10 mg/kg at day 0 | 3 7-days cycles 2% from days 6, 27, 48 | 100 days |
| *Zhang et al. (2022b)* | Mice, female, C57BL/6, 10 weeks | 1 injection i/p 10 mg/kg at day 0 | 3 7-days cycles 2% from days 6, 27, 48 | 100 days |
| **Wan et al. (2022)** | **Mice, male, Balb/C, 6 weeks** | **1 injection i/p 12.5 mg/kg at day 0** | **3 7-days cycles 3% from days 6, 27, 48** | **75 days** |
| *Hu et al. (2022)* | Mice, male, Balb/C, 6 weeks | 1 injection i/p 10 mg/kg at day 0 | 3 5-days cycles 2% from days 6, 27, 48 | 68 days |
| *Cariello et al. (2022)* | Mice, male, C57BL/6J, 8 weeks | 1 injection i/p 12 mg/kg at day 0 | 3 5-days cycles 3.5% from days 5, 24, 42 | 55 days |
| *Zhang et al. (2022c)* | Mice, sex?, C57BL/6, 6 weeks | 1 injection i/p 10 mg/kg at day 0 | 3 7-days cycles 2%, 1.5% and 1.5% from days 5, 26, 47, respectively | 80 days |
| **Naydenov et al. (2022)** | **Mice, male and female, C57BL/6J, 8–10 weeks** | **1 injection i/p 10 mg/kg at day 0** | **3 5-days cycles 2.5% from days 4, 23, 42** | **60 days** |

**Table 1** (*continued*)

| Reference | Animals | AOM | DSS | Experiment duration |
|---|---|---|---|---|
| *Mesas et al. (2022)* | Mice, female, C57BL/6, 6 weeks | 1 injection i/p 12 mg/kg at day 0 | 3 5-days cycles 2%, 2% and 3% from days 5, 26, 47, respectively | 116 days |
| *Marie et al. (2022)* | **Mice, male and female, C57BL/6J, 9–10 weeks** | **1 injection i/p 10 mg/kg at day 0** | **3 5-days cycles 4% from days 1, 22, 43** | **100 days** |
| *Thanki et al. (2022)* | **Mice, sex?, C57BL/J6 129SvEv, age?** | **1 injection i/p 10 mg/kg at day 0** | **3 7-days cycles 2% from days 1, 22, 43** | **80 days** |
| *Heichler et al. (2022)* | Mice, sex?, line?, age? | 1 injection i/p ? mg/kg at day 0 | 3 7-days cycles ?% from days 1, 22, 43 | 63 days |
| *Wang et al. (2022d)* | **Mice, male, C57BL/6, age?** | **1 injection i/p 10 mg/kg at day 0** | **3 7-days cycles 2% from days 1, 22, 43** | **70 days** |
| *Wang et al. (2022c)* | Mice, female, BALB/c и C57BL/6, 4–6 weeks | 1 injection i/p ? mg/kg at day 0 | 3 7-days cycles 2.5% from days 1, 22, 43 | 70 days |
| *Zuo et al. (2023)* | Mice, male, line?, age? | 1 injection i/p 10 mg/kg at day 0 | 3 8-days cycles 2% from days 1, 23, 45 | 67 days |
| *Liu et al. (2022f)* | **Mice, male, Balb/C, 5 weeks** | **1 injection i/p 10 mg/kg at day 0** | **1 4-days cycle 2% from day 1** | **161 days** |
| *Tu et al. (2022)* | Mice, male, C57BL/6, 6 weeks | 1 injection i/p 10 mg/kg at day 0 | 1 7-days cycle 2% from day 8 | 63 days |
| *Song et al. (2023)* | **Mice, male and female, C57BL/6, 9 weeks** | **1 injection i/p 10 mg/kg at day 0** | **1 7-days cycle 2% from day 8** | **91 days** |
| *Andreuzzi et al. (2022)* | **Mice, sex?, C57BL/6J, 5 weeks** | **1 injection i/p 12.5 mg/kg at day 0** | **1 7-days cycle 2% from day 8** | **175 days** |
| *Cunningham et al. (2022)* | Mice, male and female, C57BL/6J, 10 weeks | 1 injection i/p 10 mg/kg at day 0 | 2 7-days cycles 2% from day 1, 22 | 56 days |
| *Kim et al. (2022)* | **Mice, male, C57BL/6J, 5 weeks** | **1 injection i/p 10 mg/kg at day 0** | **2 7-days cycles 2% from day 15, 36** | **56 days** |
| *Huang et al. (2022a)* | Mice, sex?, Balb/C, 6–8 weeks | 1 injection i/p 10 mg/kg at day 0 | 2 7-days cycles 2% from day 15, 43 | 84 days |
| *Roca-Lema et al. (2022)* | Mice, sex?, C57BL/6J, age? | 1 injection i/p 10 mg/kg at day 0 | 4 4-days cycles 1.7% from days 8, 26, 44, 62 | 80 days |
| *Guo et al. (2022)* | Mice, male, Balb/C, 4–6 weeks | 1 injection i/p 10 mg/kg at day 0 | 4 6-days cycles 3% from days 4, 24, 44, and 2.5% from 67, respectively | 107 days |
| *Liu et al. (2022a)* | Mice, sex?, C57BL/6J, age? | 1 injection i/p 10 mg/kg at day 0 | 4 7-days cycles 1.25% from days 8, 29, 50, 71 | 91 days |
| *Li et al. (2022b)* | Mice, male, линия?, 10 weeks | 1 injection i/p 10 mg/kg at day 0 | 4 7-days cycles 1.6% from days 8, 29, 50, 71 | 91 days |
| *Chang et al. (2022)* | Mice, male, C57BL/6J, 8 weeks | 1 injection i/p 10 mg/kg at day 0 | 4 7-days cycles 2% from days 8, 22, 36, 50 | 100 days |
| *Huang et al. (2022b)* | **Mice, male, C57BL/6, 6 weeks** | **1 injection i/p 12.5 mg/kg at day 0** | **4 7-days cycles 2.5% from days 8, 29, 50, 71** | **119 days** |
| *El-Hindi et al. (2022)* | Mice, male and female, line?, 9 weeks | (total 10 mg/kg) 2 injections i/p 5 mg/kg at 0 and 1 days | 3 5-days cycles 2% from days 2, 22, 42 | 84 days |

**Table 1** (*continued*)

| Reference | Animals | AOM | DSS | Experiment duration |
|---|---|---|---|---|
| *Vega et al. (2022)* | Mice, sex?, C57BL/6J, 10 weeks | (total 20 mg/kg) 2 injections i/p 10 mg/kg at 0 and 7 days | 2 7-days cycles 2% from days 6, 43 | 84 days |
| *Cai et al. (2022)* | **Mice, sex?, C57BL/6, age?** | **(total 25 mg/kg) 2 injections i/p 12.5 mg/kg at 0 and 65 days** | **4 7-days cycles 2.5% from days 6, 27, 48, 71** | **91 days** |
| *Javadzadeh et al. (2022)* | Mice, male, Balb/C, 6–8 weeks | (total 26.5 mg/kg) 3 injections i/p 15 mg/kg at 0 day, 7.5 mg/kg at 21 day and 4 mg/kg at 50 day | 2 7-days cycles 2% from days 7, 28 | 80 days |
| *Seok et al. (2022)* | Mice, male, ICR, 4 weeks | (total 30 mg/kg) 3 injections i/p 10 mg/kg at 0, 7, 14 days | 1 7-days cycle 2% from day 15 | 42 days |
| *Yao et al. (2022)* | Mice, male, C57BL/6, 11 weeks | (total 30 mg/kg) 3 injections i/p 10 mg/kg at 0, 28, 49 days | 3 7-days cycles 2.5% from days 7, 35, 56 | 63 days |
| *Wang et al. (2022e)* | Mice, female, BALB/c, age? | (total 60 mg/kg) 5 injections i/p 12 mg/kg at 0–4 days | 3 7-days cycles 2% from days 5, 27, 47 | 67 days |
| *Tajasuwan et al. (2022)* | **Rats, male, Wistar, 5 weeks** | **(total 30 mg/kg) 2 injections subcutaneously 15 mg/kg at 0 and 7 days** | **2 7-days cycles 4% from days 14, 28** | **70 days** |

**Notes.**

Color marked works in which the development of adenocarcinomas is noted.

Bold indicates the development of adenocarcinomas.

were detected adenocarcinoma on day 84 of the experiment (12 weeks) after a single intraperitoneal injection of 12.5 mg/kg AOM and three cycles for 7 days 2.5% DSS.

Thus, the AOM/DSS-induced CAC model is widely used to study carcinogenesis associated with chronic inflammation of the large intestine. However, the model is not standardized: in different studies, the doses and duration of exposure to AOM and DSS vary significantly, which makes it difficult to analyze and compare the obtained experimental data. Moreover, studies are carried out on mice of different lines, different sexes, which are not indicated in many articles (Table 1). Differences in the results obtained, in addition to the above, may be due to differences in the microbiome. The gut microbiota plays role in beneficial functioning for the organism cells, including protection from pathogen colonization and forming the immune cell interaction, and therefore, its disruption, termed dysbiosis, is associated with the risk of intestinal inflammation and CAC (*Richard et al., 2018*). It appeared, studies demonstrated that chronic inflammation induced by AOM/DSS treatment leads to gut dysbiosis in SPF mice. In this CAC model, bacterial diversity in the gut is significantly decreased and subsequently, the mice develop tumors in the gut. Nevertheless, antibiotic administration can reverse this phenotype. Consistently, APC$^{Min}$ CAC mice develop fewer tumors when housed in germ-free conditions (*Dove et al., 1997*). Moreover, gnotobiotic mice colonized with microbiota derived from CAC mice have an increased tumor burden and incidence in comparison to those colonized with microbiota derived from healthy mice (*Zackular et al., 2013*). Vancomycin use in AOM/DSS mice

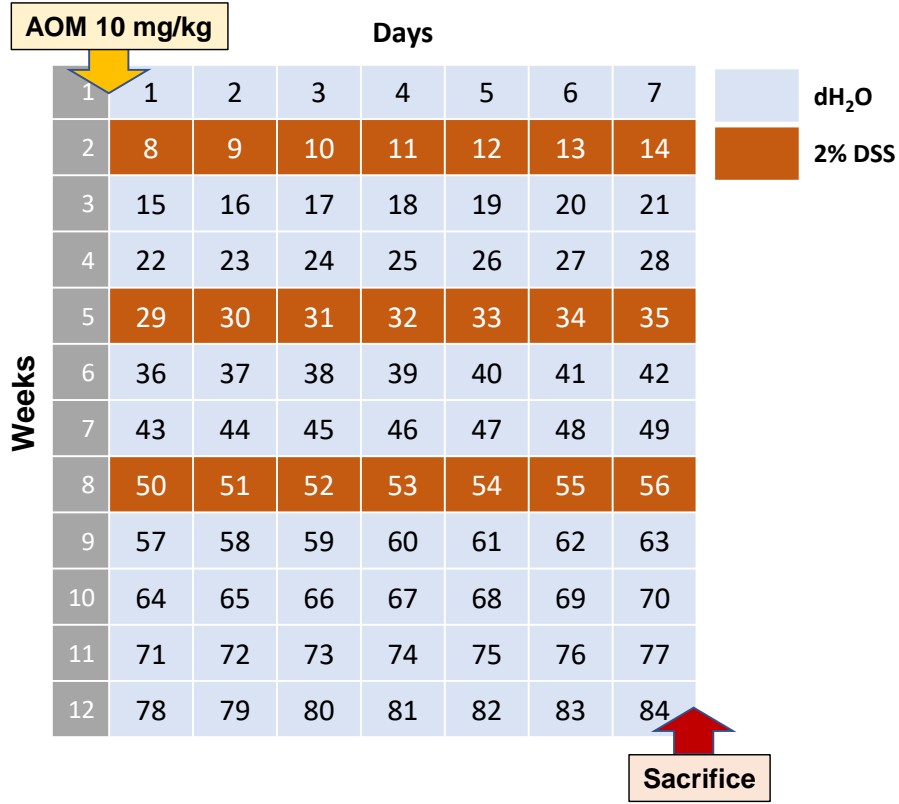

**Figure 1  Most commonly used AOM/DSS CAC induction regimen.**

reduces tumor development by the loss of neutrophils that induced DNA damage in the intestinal epithelial cells (*Tanaka, Ito & Isobe, 2016*). These investigations strongly suggest that inflammation-driven tumorigenesis is dependent on the gut microbiota. The significance of the microbiota in the development of CAC is discussed in detail in recent reviews (*Leystra & Clapper, 2019*; *Nagao-Kitamoto, Kitamoto & Kamada, 2022*).

Moreover, differences may be related to individual resistance to hypoxia, which plays an important role in the development of CAC. It is known that acute and chronic ulcerative colitis is more severe in susceptible to hypoxia animals (*Dzhalilova et al., 2018a*; *Dzhalilova et al., 2018b*). In addition, features of glioblastoma progression depending on individual resistance to hypoxia were found (*Dzhalilova et al., 2023*). However, the features of the course of CRC depending on individual resistance to hypoxia on the AOM/DSS model have not yet been studied.

Another shortcoming of this model should be noted. There are important differences in invasive and metastatic potentials of CRC between animal models and human disease. At diagnosis, at about 50% of CRC patients have lymphatic metastases and 33% have hematogenous metastases (*Tanaka, 2009*). However, the frequency of metastases is extremely low in AOM models of CRC (*Rosenberg, Giardina & Tanaka, 2009*). In human CRC, the disease progresses in order, and a common route of hematogenous metastasis

first reaches the liver, and subsequently lungs. Nonetheless, metastatic liver tumors are rare in the AOM models of rodent CRC. A recent study (*Tanimura et al., 2021*) demonstrated that in comparison with AOM/DSS mice at 10 and 20 weeks, submucosal tumor infiltration and tumor invasion into vessels were markedly increased at 30 weeks, *i.e.,* a longer duration of the experiment is required. Furthermore, it was demonstrated that an AOM/DSS model with a low dose of DSS (1%) can be used to reliably induce colorectal carcinogenesis measured as preneoplastic lesions in both B6 and A/J mouse strains while limiting severe symptoms. This study emphasizes the importance of adjusting the treatment regimen according to mice strain and study aims in future investigations employing the AOM/DSS model (*Arnesen, Müller & Aleksandersen, 2021*). Therefore, although animal models of chemical-induced CRCs have provided much information about human disease, many research approaches have not been made available, and further research to establish animal models of metastasis is needed.

## CONCLUSIONS

There are different types of CRC, differing in molecular mechanisms of occurrence and features of the course. Therefore, it is impossible to create one single model that reflects all the features of CRC. Development of spontaneous, genetically engineered, transplantation and chemically induced models allows to reproduce and study different types of CRC and certain aspects of CRC pathogenesis. Inflammatory bowel disease is an important risk factor for CRC. The AOM/DSS-induced CAC model is used to study carcinogenesis associated with chronic inflammation of the large intestine. However, this model is not standardized: according to different studies, the doses and exposure to AOM and DSS vary significantly, which makes it difficult to analyze and compare the obtained experimental data.

### Funding
This work was supported by the Russian Science Foundation grant (No. 23-25-00294). The funders had no role in study design, data collection and analysis, decision to publish, or preparation of the manuscript.

### Grant Disclosures
The following grant information was disclosed by the authors:
Russian Science Foundation grant: 23-25-00294.

### Competing Interests
The authors declare there are no competing interests.

### Author Contributions
- Dzhuliia Dzhalilova conceived and designed the experiments, performed the experiments, analyzed the data, authored or reviewed drafts of the article, and approved the final draft.

- Natalia Zolotova conceived and designed the experiments, performed the experiments, analyzed the data, prepared figures and/or tables, and approved the final draft.
- Nikolai Fokichev analyzed the data, authored or reviewed drafts of the article, and approved the final draft.
- Olga Makarova analyzed the data, authored or reviewed drafts of the article, and approved the final draft.

### Data Availability

This is a literature review.

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
