# Peer review of "Murine models of colorectal cancer: the azoxymethane (AOM)/dextran sulfate sodium (DSS) model of colitis-associated cancer"

_PeerJ, doi:10.7717/peerj.16159_

## Round 0.1 · original submission · Major Revisions

Dear authors,

First of all, we apologize for the delay in the review process.

As you will note the reviewers are very positive about your work, however, some points deserve a careful revision.

Please address all the review points in your point-by-point rebuttal letter.

Thank you for submitting your work to PeerJ.

**Language Note:** PeerJ staff have identified that the English language needs to be improved. When you prepare your next revision, please either (i) have a colleague who is proficient in English and familiar with the subject matter review your manuscript, or (ii) contact a professional editing service to review your manuscript. PeerJ can provide language editing services - you can contact us at copyediting@peerj.com for pricing (be sure to provide your manuscript number and title). – PeerJ Staff

Reviewer 1 ·

Basic reporting

In the manuscript "Murine models of colorectal cancer. AOM/DSS model of colitis-associated cancer. A Literature Review." the authors review the existing murine models of CRC with a focus on the models of colitis-associated CRC. The manuscript could be relevant for experimental biologists and oncologists for help in the research of CRC. This is an interesting study; the paper is generally well-written and structured. However, in the introduction is necessary to include more references from original papers instead of citing numerous reviews to demonstrate how the work fits into the broader field of knowledge.
in Page 16, lines 421 to 423 some references are missing.
The Figure and table are relevant to the article's content and are appropriately described and labeled.
Although this topic has been reviewed on other occasions, the authors present an updated summary of the use of these experimental models.

Experimental design

For the study, the authors checked PubMed and Google from 01/2018 to 05/2023 for reviews of CRC models. For articles using the AOM/DSS CRC model, they checked in PubMed from 01/2022 to 01/2023. However, it is unclear why the AOM/DSS CRC model selected this date instead of the last two or three years. This is a widely used model. Selecting the last year's articles, particularly those published immediately after the covid 19 pandemic, will leave out valuable information. Additionally, readers need to know the molecular weight of the DSS that is used in most studies.

Validity of the findings

The authors described the advantages and disadvantages of the models. The authors claim that the "AOM/DSS model is not standardized, which makes it difficult to analyze and compare data from different studies." The authors should discuss posible explanations about this differences. Different microbiota could be an explanation. Also, it could be helpful if the authors wrote a subsection describing the main findings that have been made from this model (for example, the description of signaling pathways, the participation of cells of the immune system, the use of anti-cancer drugs, etc.).
Additionally, exposure to AOM/DSS can induce the formation of a complete process of colon oncogenesis, progressing from the initial proliferation of crypts to the final development of colon carcinoma. Since multiple studies have reported early and late histologic changes in this model, creating a table indicating the changes expected to be found in this model in the early and late stages of carcinogenesis could be helpful.

Additional comments

no comment

Reviewer 2 ·

Basic reporting

Overall, Dzhalilova et al. have written an interesting review that deserves publication. The basic reporting is good, despite typographical and grammatical errors. Occasionally the authors misuse an important word, certain sections of the review need better referencing, even though they are not the major focus of the manuscript. The study’s strong point and its emphasis is on AOM/DSS model of colitis-associated colorectal cancer in murine models. It was enjoyable to read, thank you for sending it to me for review.
Point 1: Lines 52-54
“Comprehensive genomic analysis demonstrated that CRC is characterized by molecular heterogeneity, its individual cases are unique and contain, on average, 76 non-silent mutations each” needs referencing.

Point 2: Lines 54-74 also need referencing- see Guinney et al. Nat. Med. 21, 1350-1356. https://doi.org/10.1038/nm.3967

Point 3: Lines 77-80 “The risk factors for CRC are considered to be such as genetic predisposition, high body 78 mass index, lack of physical activity, tobacco and alcohol use, a diet high in red meat, fried 79 foods, low dietary fiber, vitamin D deficiency, use of poor-quality drinking water, etc.”
omit “considered to be such as”
Point 4: Lines 90-93 “Due to the remaining unclear etiology and insufficient understanding of the pathogenesis of IBD, methods for their prevention, diagnosis and treatment are not enough effective.
rearrange wording to:
Due to unclear etiology and insufficient understanding of IBD pathogenesis, methods for prevention, diagnosis and treatment are not effective enough.
Point 5: Lines 130-132
Studies of biopsy material provide information only about CM changes from several local areas of the colon.
Does CM refer to CMS? Either give explanation of the acronym “CM” or change to CMS.
Point 6: Line 144 “The query “colorecral cancer model" was used...”
The misspelling of colorectal cancer needs correction.
Point 7: Lines 172-4
“According Ye et al. (2020) KRAS, KIT, PIK3CA, MET and EGFR are among the most frequently mutated driver genes, and TP53, APC, CDKN2A, STK11 and FBXW7 are the most frequently mutated tumor suppressor genes in patients with CRC”.
Change “driver” to “oncogene”, the word “driver” includes both oncogenes and tumor suppressors, also add “ to” ... according to Ye et al. “...
Point 8: Lines 179-181
“The offspring of C57BL/6 180 mice treated with the N-ethyl-N-nitrosourea mutagen developed multiple adenomas throughout the intestine at an early age (Moser, Pitot & Dove, 1990)”.
This is a little misleading the way it is written, not all animals treated with the carcinogen had the phenotype. One animal had multiple intestinal neoplasia, that animal was used to establish the Apc Min strain. Would be best to clarify the statement.
Point 9: Lines 193-196
“The APCMin mouse model is the only animal model of CRC in which a single mutation leads to the development of an invasive organ-specific intestinal tumor with a typical CRC progression from adenoma to adenocarcinoma (Nascimento-GonCalves et al., 2021)”.
The exact wording from the reference is: “The Min mouse model is the only animal model of cancer that contains a single genetic alteration capable of producing a fully penetrating, consistent, and organ-specific tumor phenotype”.
It is important to clarify that the Apc Min mouse leads to adenomas of the murine small intestine, that only a small proportion of the animals develop adenocarcinomas and do so at a late age due to further mutation. It is necessary to point out that, unlike the mouse model, humans very rarely get tumors of the small intestine.
Point 10: Line 206, “wish” should read “with”.
Point 11: Lines 212-214: “It is caused by autosomal dominant mutations in the DNA mismatch repair genes such as MLH1, MSH2, MSH6, and PMS2, that leads to the development of a variety of cancers, including colorectal cancerDeveloped tumors…”.
There is a grammatical error in punctuation that needs correcting.
Point 12: Lines 219-221
“Therefore, a number of mouse models were developed with knockouts MMR genes such as Mlh1, Mlh3, Msh2, Msh6, and Pms2. For example, Mlh1 and Msh2 knockout mice develop tumors predominantly in the small intestine”.
The authors need to be more specific here about knockouts and MMR genes-
Mice with a null knockout of the Mlh1 gene develop intestinal tumors and additionally cause sterility of both male and female mice by failure to progress beyond meiotic prophase and arrest at pachytene. Null knockout of Msh2 alone in mice lead to the development of lymphomas, conditional knockout of Msh2 alone under the control of a constitutive Villin-Cre permit routine development of adenocarcinomas of the small intestine without lymphoma.
In support see (not required):
Edelmann, W., Cohen, P.E., Kane, M., Lau, K., Morrow, B., Bennett, S., Umar, A., Kunkel, T., Cattoretti, G., Chaganti, R., et al. (1996). Meiotic pachytene arrest in MLH1-deficient mice. Cell 85, 1125–1134. https://doi.org/10. 1016/s0092-8674(00)81312-4.
Reitmair, A.H., Schmits, R., Ewel, A., Bapat, B., Redston, M., Mitri, A., Waterhouse, P., Mittru¨ cker, H.W., Wakeham, A., Liu, B., et al. (1995). MSH2 deficient mice are viable and susceptible to lymphoid tumours. Nat. Genet. 11, 64–70. https://doi.org/10.1038/ ng0995-64. 143. Siegel, E.C., and Bryson, V
Kucherlapati, M.H., Lee, K., Nguyen, A.A., Clark, A.B., Hou, H., Jr., Rosulek, A., Li, H., Yang, K., Fan, K., Lipkin, M., et al. (2010). An Msh2 conditional knockout mouse for studying intestinal cancer and testing anticancer agents. Gastroenterology 138, 993–1002.e1. https://doi.org/10.1053/j. gastro.2009.11.009.
Point 13: Lines 226-7 “Aimals wish p53 knockout ...”. Should read “Animals with p53 knockout...
Point 14: Lines 227-229 “However, the association of APCMin and p53 knockout mutations promoted an increase in aberrant crypts foci number when compared to APCMin animals (DE-Souza & Costa-Casagrande, 2018)”.
This reviewer is under the impression this statement is in error, De-souza et al. (2018) quote Reed et al. (2008) from the AR Clarke lab in their Table regarding APCMin and p53 compound mutations.
AR. Clarke states (1995) “Surprisingly, given the proposed role for loss of function mutations of the p53 gene in the development of human colorectal cancer, we have found no evidence for either an increase in the rate of adenoma formation in APC +/-, p53 -/- animals, or an increased rate of progression to malignancy compared with APC +/- p53 +/+ mice”.
AR Clarke was supported by: Fazeli, A., Steen, R.G., Dickinson, S.L., Bautista, D., Dietrich, W.F., Bronson, R.T., Bresalier, R.S., Lander, E.S., Costa, J., and Weinberg, R.A. (1997). Effects of p53 mutations on apoptosis in mouse intestinal and human colonic adenomas. Proc. Natl. Acad. Sci. USA 94, 10199–10204. https://doi. org/10.1073/pnas.94.19.10199.
Point 15: Line 233 “...found in more than 40% of patients wish CRC...”. Change “wish” to “with”.
Point 16: Line 291 “At the moment, a wide range of chemically-induced CRC models was developed”,
should read “At the moment, a wide range of chemically induced CRC models are available.”
Point 17: Line 422 ” 2,4,6-rinitrobenzenesulfonic acid (TNBS)” probably should be 2,4,6-trinitrobenzenesulfonic acid.

Experimental design

Study design is fine.

Validity of the findings

There is a benefit to the literature in publishing the review.

Additional comments

I recommend publishing after revision.

Reviewer 3 ·

Basic reporting

In my opinion it’s a interesting review about murine models of colorectal cancer. The table and figure are also interesting

Experimental design

No coment

Validity of the findings

No coment

Additional comments

In my opinion this article should be accepted. The theme is not a novelty but the data apresentation is interesting and clear

---

## Round 0.2 · accepted · Accept

Dear authors,

thank you for revising your manuscript in accordance with the Reviewers' suggestions. Your manuscript is now accepted for publication

Congratulations

Reviewer 1 ·

Basic reporting

The authors have answered all of my questions

Experimental design

The Authors have addressed all of my concerns with the original manuscript. The revised manuscript is ready for publication.

Validity of the findings

The Authors have addressed all of my concerns with the original manuscript. The revised manuscript is ready for publication.

Additional comments

The Authors have addressed all of my concerns with the original manuscript. The revised manuscript is ready for publication.

Reviewer 2 ·

Basic reporting

Dear Reviewer,
Thank you for the hard work with our article.
We very appreciate your comments, remarks and notes and the spent time for the manuscript reviewing. It goes without saying, they helped to improve our work and made it possible to pay attention to the drawbacks. We tried to take them into account in case to fix the mentioned points and hope that this will make it possible to publish our article.
Please let us answer to your comments in details:

Reviewer Point 1: Lines 52-54
“Comprehensive genomic analysis demonstrated that CRC is characterized by molecular heterogeneity, its individual cases are unique and contain, on average, 76 non-silent mutations each” needs referencing.

Author Response 1: Thank you, we added the reference. OK

Reviewer Point 2: Lines 54-74 also need referencing- see Guinney et al. Nat. Med. 21, 1350-1356. https://doi.org/10.1038/nm.3967

Author Response 2: Thank you, we added the reference. OK

Reviewer Point 3: Lines 77-80 “The risk factors for CRC are considered to be such as genetic predisposition, high body 78 mass index, lack of physical activity, tobacco and alcohol use, a diet high in red meat, fried 79 foods, low dietary fiber, vitamin D deficiency, use of poor-quality drinking water, etc.”
omit “considered to be such as”

Author Response 3: Corrected. OK

Reviewer Point 4: Lines 90-93 “Due to the remaining unclear etiology and insufficient understanding of the pathogenesis of IBD, methods for their prevention, diagnosis and treatment are not enough effective.
rearrange wording to:
Due to unclear etiology and insufficient understanding of IBD pathogenesis, methods for prevention, diagnosis and treatment are not effective enough.

Author Response 4: Corrected. OK

Reviewer Point 5: Lines 130-132
Studies of biopsy material provide information only about CM changes from several local areas of the colon.
Does CM refer to CMS? Either give explanation of the acronym “CM” or change to CMS.

Author Response 5: Thank you for this note. CM – is colon mucosa, we’ve corrected it. OK

Reviewer Point 6: Line 144 “The query “colorecral cancer model" was used...”
The misspelling of colorectal cancer needs correction.

Author Response 6: Corrected. OK

Reviewer Point 7: Lines 172-4
“According Ye et al. (2020) KRAS, KIT, PIK3CA, MET and EGFR are among the most frequently mutated driver genes, and TP53, APC, CDKN2A, STK11 and FBXW7 are the most frequently mutated tumor suppressor genes in patients with CRC”.
Change “driver” to “oncogene”, the word “driver” includes both oncogenes and tumor suppressors, also add “ to” ... according to Ye et al. “...

Author Response 7: Corrected. OK

Reviewer Point 8: Lines 179-181
“The offspring of C57BL/6 180 mice treated with the N-ethyl-N-nitrosourea mutagen developed multiple adenomas throughout the intestine at an early age (Moser, Pitot & Dove, 1990)”.
This is a little misleading the way it is written, not all animals treated with the carcinogen had the phenotype. One animal had multiple intestinal neoplasia, that animal was used to establish the Apc Min strain. Would be best to clarify the statement.

Author Response 8: Thank you, we’ve corrected it. OK

Reviewer Point 9: Lines 193-196
“The APCMin mouse model is the only animal model of CRC in which a single mutation leads to the development of an invasive organ-specific intestinal tumor with a typical CRC progression from adenoma to adenocarcinoma (Nascimento-GonCalves et al., 2021)”.
The exact wording from the reference is: “The Min mouse model is the only animal model of cancer that contains a single genetic alteration capable of producing a fully penetrating, consistent, and organ-specific tumor phenotype”.
It is important to clarify that the Apc Min mouse leads to adenomas of the murine small intestine, that only a small proportion of the animals develop adenocarcinomas and do so at a late age due to further mutation. It is necessary to point out that, unlike the mouse model, humans very rarely get tumors of the small intestine.

Author Response 9: Thank you, we’ve corrected it. OK

Reviewer Point 10: Line 206, “wish” should read “with”.

Author Response 10: Corrected. OK

Reviewer Point 11: Lines 212-214: “It is caused by autosomal dominant mutations in the DNA mismatch repair genes such as MLH1, MSH2, MSH6, and PMS2, that leads to the development of a variety of cancers, including colorectal cancerDeveloped tumors…”.
There is a grammatical error in punctuation that needs correcting.

Author Response 11: Thank you, we’ve corrected it.

Reviewer Point 12: Lines 219-221
“Therefore, a number of mouse models were developed with knockouts MMR genes such as Mlh1, Mlh3, Msh2, Msh6, and Pms2. For example, Mlh1 and Msh2 knockout mice develop tumors predominantly in the small intestine”.
The authors need to be more specific here about knockouts and MMR genes-
Mice with a null knockout of the Mlh1 gene develop intestinal tumors and additionally cause sterility of both male and female mice by failure to progress beyond meiotic prophase and arrest at pachytene. Null knockout of Msh2 alone in mice lead to the development of lymphomas, conditional knockout of Msh2 alone under the control of a constitutive Villin-Cre permit routine development of adenocarcinomas of the small intestine without lymphoma.
In support see (not required):
Edelmann, W., Cohen, P.E., Kane, M., Lau, K., Morrow, B., Bennett, S., Umar, A., Kunkel, T., Cattoretti, G., Chaganti, R., et al. (1996). Meiotic pachytene arrest in MLH1-deficient mice. Cell 85, 1125–1134. https://doi.org/10. 1016/s0092-8674(00)81312-4.
Reitmair, A.H., Schmits, R., Ewel, A., Bapat, B., Redston, M., Mitri, A., Waterhouse, P., Mittru¨ cker, H.W., Wakeham, A., Liu, B., et al. (1995). MSH2 deficient mice are viable and susceptible to lymphoid tumours. Nat. Genet. 11, 64–70. https://doi.org/10.1038/ ng0995-64. 143. Siegel, E.C., and Bryson, V
Kucherlapati, M.H., Lee, K., Nguyen, A.A., Clark, A.B., Hou, H., Jr., Rosulek, A., Li, H., Yang, K., Fan, K., Lipkin, M., et al. (2010). An Msh2 conditional knockout mouse for studying intestinal cancer and testing anticancer agents. Gastroenterology 138, 993–1002.e1. https://doi.org/10.1053/j. gastro.2009.11.009.

Author Response 12: Thank you, we searched for more details on this and found a detailed review (doi: 10.1016/j.dnarep.2015.11.015), so we removed the irrelevant information and referenced it. This is fine, reference OK.

Reviewer Point 13: Lines 226-7 “Aimals wish p53 knockout ...”. Should read “Animals with p53 knockout...

Author Response 13: Thank you very much. We’ve corrected it. OK

Reviewer Point 14: Lines 227-229 “However, the association of APCMin and p53 knockout mutations promoted an increase in aberrant crypts foci number when compared to APCMin animals (DE-Souza & Costa-Casagrande, 2018)”.
This reviewer is under the impression this statement is in error, De-souza et al. (2018) quote Reed et al. (2008) from the AR Clarke lab in their Table regarding APCMin and p53 compound mutations.
AR. Clarke states (1995) “Surprisingly, given the proposed role for loss of function mutations of the p53 gene in the development of human colorectal cancer, we have found no evidence for either an increase in the rate of adenoma formation in APC +/-, p53 -/- animals, or an increased rate of progression to malignancy compared with APC +/- p53 +/+ mice”.
AR Clarke was supported by: Fazeli, A., Steen, R.G., Dickinson, S.L., Bautista, D., Dietrich, W.F., Bronson, R.T., Bresalier, R.S., Lander, E.S., Costa, J., and Weinberg, R.A. (1997). Effects of p53 mutations on apoptosis in mouse intestinal and human colonic adenomas. Proc. Natl. Acad. Sci. USA 94, 10199–10204. https://doi. org/10.1073/pnas.94.19.10199.

Author Response 14: Thank you, we’ve corrected it. OK, (TP53 seems to be involved in CRC metastasis when APC/TP53/KRAS are all altered).

Reviewer Point 15: Line 233 “...found in more than 40% of patients wish CRC...”. Change “wish” to “with”.

Author Response 15: Corrected. OK

Reviewer Point 16: Line 291 “At the moment, a wide range of chemically-induced CRC models was developed”, should read “At the moment, a wide range of chemically induced CRC models are available.”

Author Response 16: Corrected. OK

Reviewer Point 17: Line 422 ” 2,4,6-rinitrobenzenesulfonic acid (TNBS)” probably should be 2,4,6-trinitrobenzenesulfonic acid.

Author Response 17: Corrected. OK

Once again we want to thank you for your attention! You are most welcome-

Experimental design

Srudy design is fine.

Validity of the findings

Not applicable for review.

Additional comments

The authors have corrected all concerns asked after reading the first submission of the manuscript.